# Relationship between Burden, Quality of Life and Difficulties of Informal Primary Caregivers in the Context of the COVID-19 Pandemic: Analysis of the Contributions of Public Policies

**DOI:** 10.3390/ijerph20065205

**Published:** 2023-03-15

**Authors:** Tania Gaspar, Marta Raimundo, Sofia Borges de Sousa, Marta Barata, Tulia Cabrita

**Affiliations:** 1SPIC, Hei-Lab, Lusófona University, 1749-024 Lisbon, Portugal; 2ISAMB, Medicine Faculty, Lisbon University, 1649-026 Lisbon, Portugal; 3SPIC, Psychology and Life Sciences School, Lusófona University, 1749-024 Lisbon, Portugal; 4Aventura Social Associação, 1649-026 Lisbon, Portugal; 5CLISSIS, Psychology Institute, Lusiada University, 1349-001 Lisbon, Portugal

**Keywords:** caregivers, difficulties, public policies, mental health

## Abstract

The study aimed to characterize and understand the difficulties experienced by informal caregivers from a bio-psychosocial and environmental perspective, taking into account the socio-demographic and health characteristics of the informal caregiver and the person cared for, quality of life, perceived burden, social support, and the impact of the COVID-19 pandemic on the informal caregiver and the person cared for. The participants were 371 informal primary caregivers, 80.9% female, aged between 25 and 85 years, mean 53.17 (SD = 11.45) years. Only 16.4% of the informal caregivers benefited from monitoring and training for informal caregiver skills; 34.8% received information on the rights of the person being cared for; 7.8% received advice or guidance on the rights and duties of the informal caregiver; 11.9% of the caregivers benefited from psychological support; and 5.7% participated in self-help groups. A convenience sample was used, and data were collected via an online questionnaire. The main findings show that the major difficulties experienced by caregivers are related to social constraints, the demands of caring, and the reactions of the person cared for. The results reveal that the burden of the main informal caregivers is explained by the level of education, quality of life, level of dependence of the person cared for, level of difficulties, and social support. The COVID-19 pandemic impacted caregiving by increasing the perceived difficulty of accessing support services, such as consultations, services, and support; causing distress feelings in the caregiver, such as, anxiety and worry; increasing the needs and symptoms of the person cared for; and increasing the degree of isolation, for both, the informal caregiver and the person cared for.

## 1. Introduction

The world is facing several socio-demographic changes, such as an increasing average life expectancy and increasing diseases, such as chronic diseases [1]. It is estimated that about 703 million people (9%) are over 65 years old [2], and about 1 billion people (15%) worldwide have a disability [3,4,5]. In Portugal, there is an increase in the number of people aged over 65 from 20% in 2015 to 23.4% in 2021 [6,7]. The increase in aging and other health problems is often accompanied by an increase in limitations at the level of the individual’s functionality and ability to perform activities of daily living. These limitations can lead to the loss of autonomy, increased dependence, and the need for care from others [1,8,9,10,11]. Thus, it can be seen that this increase in the elderly population and changes in the composition of households are also reflected in the number of informal caregivers [1]. In Portugal, it is estimated that about 1 million people (10% of the population) play the role of informal caregivers, mostly performed by women (692,305). An informal caregiver is defined as the spouse or unmarried partner, relative, or kin up to the 4th degree in the direct or collateral line of the person being cared for (e.g., children, grandchildren, great-grandchildren, great-great-grandchildren, siblings, parents, uncles, grandparents, great-grandparents, great-great-uncles, or cousins) [12].

Informal care is often provided by individuals close to the dependent person who, in most cases, are family members or individuals with whom the dependent person maintains an affective bond. Caregivers perform various services for which they have no training and receive no type of remuneration [13,14,15,16]. The role of informal caregiver is mostly played by women, middle-aged individuals, unemployed or domestic workers, religious people, and individuals with some health problems [17,18].

The informal caregiver performs several tasks (partially or fully) and has several responsibilities, seeking to provide well-being and quality of life to the dependent individual [19]. They are primarily responsible for keeping the dependent individual at home and thus avoiding institutionalization. The tasks performed by caregivers are at the level of personal care, such as feeding and hygiene, assistance in moving and transferring the dependent person, housekeeping, assistance in preparing and taking medication, wound care, and monitoring health care equipment, among others [1,20,21,22]. The amount of time spent providing care is associated with the number of tasks performed by the caregiver [22].

The provision of care is associated with numerous challenges that affect informal caregivers at a personal, professional, and social level, leading to high physical and psychological stress, which is reflected in their health, well-being, and quality of life [1,17,18,23,24,25]. Being a caregiver is associated with decreased quality of life and well-being, high levels of depression, greater financial burden, greater impact at the physical level, reduced quality of interpersonal relationships and leisure time, and increased burnout [19,26,27,28,29,30]. Furthermore, being a caregiver is associated with an increased risk of having health problems, both physical and mental, due to the high demands they are subjected to. This risk increases in female caregivers, caregivers who have received training (which is often not sufficient for the care they are providing), and caregivers who have been performing this role for at least two years [17]. Studies report that informal caregivers may have a greater tendency to neglect their health due to the time they devote to caregiving responsibilities [17,22,28].

The caregivers’ quality of life is associated with several factors, namely their health status (physical and psychological), level of burden, financial issues, social support, health conditions, and degree of dependence on the caregiver [17,21,28,29]. These well-being and mental health consequences affect female caregivers, intensive informal caregivers (more than 11 h of caregiving per week) more [18,19,23], and married caregivers [23].

The literature shows that, despite the difficulties experienced by the caregivers, their perceptions of burden and subjective well-being vary, even when they are faced with similar situations [29]. Caregiver burden occurs when the caregiver evaluates care provision as something negative and stressful [31]. It is influenced by the emotional, social, and financial stress imposed by the specificities of the person cared for [29,30]. Caregivers’ burden is associated with several factors such as the duration of care, the amount of care provided per day, and the level of dependence on the caregiver [28,31]. With regard to the latter, it is associated either with physical dependence or with cognitive or behavioral aspects. The level of dependence from neurodegenerative disorders, some oncological diseases, physical disabilities, and the presence of comorbidities is associated with high caregiver burden [31]. In addition, spouse caregivers report a higher emotional burden since they see care provision as an obligation towards their spouse [28]. Caregivers’ perceived burden impacts their quality of life and activities of daily living [22,28,32], with quality of life being a predictor of caregiver depression and anxiety [28]. When high levels of depression occur, there is a greater propensity for the abandonment of the caregiver role to occur [19].

However, if, as mentioned so far, caregiving may represent a stressful situation and entail negative consequences for the caregiver, on the other hand, it may represent benefits for the caregiver, such as the perception of a positive feeling of reward or a closer relationship with the individual being cared for [18], which may lead to an increase in life satisfaction and, consequently, a reduction in depressive feelings [19].

Social support plays an important role in improving quality of life and reducing the perception of burden [21]. By supporting the informal caregiver and contributing to improving their health, social support provides better care conditions for the dependent person. It is important to increase the social support networks, either through the caregivers’ close relationships or through health professionals, in order to reduce the burden felt by caregivers, while seeking to increase their well-being and quality of life [33], which are associated with the quality of care provided by the informal caregiver. Thus, the caregiver is an important part of care provision, and it is essential to support caregivers in managing their difficulties [1,21].

The COVID-19 pandemic, the respective isolation measures, and the difficulties in accessing health and social services that were focused exclusively on fighting the pandemic at the time had a greater impact on certain groups, namely the elderly, people with chronic illnesses, people with socio-economic difficulties, women, and the unemployed. Informal caregivers and their cared-for persons were one of the groups most affected by the pandemic and its restrictions [34,35,36].

The informal caregivers were extremely important for stress management and psychological well-being promotion during the pandemic when people’s mobility and everyday activities have been considerably influenced by lockdown measures and other containment interventions. Vulnerable groups (such as older people and low-income people) were highly dependent on informal caregivers. And informal caregivers have contributed to COVID-19 containment and recovery [37].

The main objective is to characterize and understand the difficulties experienced by informal caregivers in the context of the COVID-19 pandemic from a bio-psychosocial and environmental perspective. The difficulties experienced by informal caregivers were considered in terms of relational problems (RP), social constraints (SR), care demands (CE), reactions to caregiving (RC), family support (F), and professional support. We aimed to characterize these difficulties from a bio-psychosocial and environmental perspective, taking into account the socio-demographic and health characteristics of the informal caregiver and the person cared for, quality of life, perceived burden, social support, and the impact of the COVID-19 pandemic on the informal caregiver and the person cared for. The following research questions can be asked: Will the informal caregiver’s sociodemographic characteristics, health, quality of life, and social support influence his/her perception of difficulties as a caregiver? What supports and benefits were most used by caregivers in the context of the COVID-19 pandemic?

## 2. Materials and Methods

### 2.1. Participants

This is a cross-sectional, quantitative study, with a convenience sample. A sample consisting of 371 informal primary caregivers, 80.9% female, aged between 25 and 85 years, mean of 53.17 (SD = 11.45) years.

### 2.2. Instruments

A battery of instruments was applied, including: (1) a sociodemographic questionnaire; (2) questionnaire on knowledge about the status of the informal caregiver; (3) WHOQOL-BREF Scale; (4) Zarit Burden Scale; (5) Social Support Satisfaction Scale (ESSS); (6) Caregiver Difficulties Assessment Index (CADI); and (7) Barthel Index.

#### 2.2.1. Sociodemographic Questionnaire

Designed with the purpose of collecting information on the informal caregiver’s sociodemographic data and household composition. In addition, this questionnaire included questions on care provision, namely its duration, whether the caregiver is the only person providing care to the dependent person, and whether it is the first time that he/she takes on the role of caregiver. Finally, the questionnaire included questions on the caregiver, namely his/her age, gender, degree of kinship/relationship with the caregiver, and the problem/diagnosis leading to the situation of dependence.

#### 2.2.2. Caregiver Difficulties Assessment Index (CADI) [38]

The Carers Assessment of Difficulties Index was used to assess the caregiver’s difficulties, which is the Portuguese version of the Carers Assessment of Difficulties Index (CADI) [38], and was validated for the Portuguese population by Sequeira [39].

This instrument is composed of 30 items that refer to potential difficulties associated with caring for older people. The items are answered on a Likert-type scale ranging between (1) this does not happen in my case, (2) this happens in my case and I feel that it does not disturb me, (3) this happens in my case and causes me some disturbance, and (4) this happens in my case and disturbs me a lot [36]. This author grouped the questions related to difficulties into six factors, which encompass relational problems (RP), social constraints (SR), care demands (CE), reactions to care (RC), family support (FA), and professional support (PA). The index score ranges between 30 and 120 points, and it was found that the higher the score obtained by the individual, the greater the number of difficulties associated with care provision.

With regard to the psychometric characteristics of the Caregiver Difficulties Assessment Index, it was found to have good internal consistency, with a Cronbach’s alpha (α) of 0.94. In addition, it is possible to observe that the analysis of the principal components with orthogonal rotation using the varimax method allows identifying that 63.6% of the total variance is explained by the six factors [38].

#### 2.2.3. Quality of Life Questionnaire (WHOQOL-BREF)

Quality of life was assessed using the WHOQOL-BREF scale [40], adapted and validated for the Portuguese population by Canavarro et al. [41] and Vaz Serra [42]. The WHOQOL-BREF scale consists of 26 questions, two of which consist of general QoL questions, and the remaining 24 represent each of the 24 facets that make up the original instrument. The participant answers the questions by self-reporting on a Likert scale ranging from 1 to 5, whereby the higher the score obtained, the better the QoL. Except for the first two questions, the instrument has 24 items organized into four dimensions: Physical QL (7 items), psychological QL (6 items), social QL (3 items), and environmental QL (8 items). As regards the psychometric characteristics of the Portuguese version of this instrument, we found that it has good internal consistency indices regarding the set of domains (α = 0.79) and items (α = 0.92) [41].

#### 2.2.4. Zarit Overload Scale [43]

Caregiver burden was assessed using the Zarit Caregiver Burden Interview (1983), adapted and validated for the Portuguese population by Ferreira et al. [43] This instrument aims at identifying the factors leading to caregiver burnout, measuring their health, psychological, and socioeconomic well-being, and their relationship with the dependent person.

The scale consists of 22 items on a Likert-type scale, with scores ranging from 0 (never) to 4 (almost always). Five factors were identified by the author: Loss of control (Lp), sacrifice (S), dependence (D), fear/anxiety (RA), and self-criticism (CA). The scale scores range from 0 to 80 points, and the higher the score, the higher the level of burden experienced by the patient [43].

As regards the psychometric characteristics of the scale, it was found to have good internal consistency, with a Cronbach’s alpha (α) of 0.88. The total correlation of the items ranged between 0.269 and 0.710, with all items showing statistical significance (*p* < 0.01). The factor validity of the scale through the Kaiser–Meyer–Olkin (KMO) criterion calculation (0.733) demonstrates a high correlation between the scale items. The extraction of factors through the principal component’s method with varimax rotation revealed a factorial structure that explains 60.959% of the total variability of the items [43].

#### 2.2.5. Barthel Index [44]

The Barthel Index, described and published by Mahoney and Barthel [44], was validated for the Portuguese population by Araújo et al. This is an instrument used in research and clinical practice, with the purpose of assessing the individual’s functional capacity and autonomy in performing certain activities of daily living [44]. These activities include eating, personal hygiene, using the toilet, sphincter control, bathing, dressing, and undressing, transferring from chair to bed, walking, and climbing up and down stairs. The response options vary between 0, 5, 10, and 15, depending on the item in question. The final score of the index varies between 0 (maximum dependence) and 100 (total independence). The Barthel Index was used to assess the level of dependence of the person cared for. It was completed by the informal caregiver, reporting the person cared for characteristics [45].

As regards the psychometric properties of the instrument validated for the Portuguese population, we found that it has a high level of reliability, with a Cronbach’s alpha (α) of 0.96, and that the scale items have correlations with the total scale between r = 0.66 and r = 0.93. In addition, it is possible to observe that the principal components analysis with varimax rotation confirms the unidimensional nature of the index, with all items presenting a factor loading higher than 0.71 (the magnitude of the values lies between 0.71 and 0.94) [44].

#### 2.2.6. Social Support Satisfaction Scale (ESSS) [46]

The caregivers’ social support was assessed using the Satisfaction with Social Support Scale (ESSS) of Pais-Ribeiro [46]. This scale consists of 15 items on a Likert-type scale ranging from 1 (strongly agree) to 5 (strongly disagree). The scale is composed of four factors: the factor “satisfaction with friends”, which includes five items and measures the subject’s satisfaction with his/her friendships; the factor “intimacy”, which includes four items and measures the perception of the existence of intimate social support; the factor “satisfaction with the family”, which includes three items and assesses the satisfaction with the existing family social support; and, finally, the factor “social activities”, which includes three items and measures the personal satisfaction with the social activities performed by the individual.

With regard to the psychometric characteristics of this instrument, the internal consistency of the total scale is adequate, with a Cronbach’s alpha (α) of 0.85. Regarding the internal consistency of each factor, the factor “satisfaction with friends” has a Cronbach’s alpha (α) of 0.83, the factor “intimacy” of 0.74, the factor “satisfaction with family” of 0.74, and the factor “social activities” of 0.64. The four factors explain 63.1% of the total variance of the scale [46].

#### 2.2.7. Knowledge Questionnaire on the Statute of the Informal Caregiver (Law no. 100/2019)

Knowledge about the Informal Caregiver Statute was assessed through a 22-item questionnaire built on the Informal Caregiver Statute. This included questions such as “B8. Do you have access to information that enlightens you about all the support the individual being cared for is entitled to?” or “B18. Have you ever benefited from informal caregiver support allowances?”, and the answers could vary between “Yes” and “No”, “Yes”, “No” and “Don’t know/Unknown” and “Yes”, “No”, “Don’t know/Unknown” and “Not applicable”, depending on the question asked [47].

### 2.3. Procedure

The data collection procedure consisted first of contacting several organizations and associations related to informal caregivers for convenience, such as CERCI’s, associations supporting informal caregivers, associations supporting specific groups (e.g., cerebral palsy, autism, among others) and support groups, to support the research team in disseminating the questionnaire to informal caregivers. The study was developed in the context of the COVID-19 pandemic (March 2021 to December 2021), which made it very difficult to access the study population.

Participation in the study was voluntary, and the anonymity and confidentiality of participants and the data collected were ensured throughout the study. Data were collected through an online questionnaire, and the link to access the questionnaire was shared with the participants through an email and/or support group in social networks. The participant was informed of the purpose of the study in which they would participate, and their consent was sought. After data collection, each questionnaire was assigned a number and entered into a database for later statistical analysis. The questionnaire was approved by the ethics committee of the ARSLVT/Health Ministry pro.023/CES/INV/2014.

### 2.4. Data Analysis

As regards the statistical procedures, the analysis was performed using the Statistical Package for the Social Sciences (SPSS) version 25 for Windows. The descriptive statistics analysis of the instrument dimensions and the total scores of the instruments was performed, as well as the analysis of correlations and Student’s t-test was used to analyze differences between groups. A linear regression model was calculated with informal caregiver difficulties as the dependent variable.

## 3. Results

The sociodemographic data of the participants can be observed in Table 1.

Taking into account that the study addressed the caregiver’s role, Table 2 describes the characterization of the caregiver’s role. We found that 74.4% of the participants are first-time caregivers, 81.4% are primary caregivers, and the main reason for the need for care provision is disease, with 214 participants indicating it. There is a huge lack of knowledge about the Caregiver’s Act among 86.3% of the respondents.

The data collected allow us to characterize the psychological variables under study (Table 3). Higher values were found in the dimensions of total, psychological, and physical quality of life. In the satisfaction with social support, family and friends were the most relevant.

The results of the analysis of the associations between the variables (Table 4) show significant associations between all results. Higher levels of quality of life are negatively associated with the perception of the caregiver’s burden and the perception of difficulties by the caregiver, and higher levels of quality of life are positively associated with a higher perception of social support, specifically with intimacy. The weakest, but significant, associations were positive between quality of life and family and social activities, and negative associations with the perception of dependence.

The results of the associations between the dimensions of the difficulties of caregiving are all significantly positively associated (Table 5).

The analysis of the gender differences in the difficulties in caring (Table 6) found significantly higher levels in women compared to men, with women showing higher levels in the dimensions of the perception of social constraints in the reactions to caring, professional support, and total caregiver difficulties.

In the analysis of the caregiver’s difficulties, taking into account the professional situation, we found significant differences in the perception of relational problems and professional support, with the non-active caregivers showing higher values when compared to the non-active caregivers (Table 7).

When we compare the different levels of schooling (Table 8), we find that in the dimensions of social problems and social constraints, there are higher levels in the group with higher education when compared to compulsory education.

When studying the dimensions of the difficulties perceived by the caregiver according to the perception of quality of life, significant differences were found in all dimensions, with higher values in the poor quality of life group (Table 9).

Significant differences were found in all dimensions of caregiver difficulties in relation to the perception of quality of life, with higher values in the poor health group (Table 10).

In the study of the perception of the caregiver’s difficulties, significant differences were found in the reactions to caring, with the group aged up to 50 years having higher levels when compared with the group aged over 51 years (Table 11).

The results presented indicate that the variables education (B = 0.06; *p* = 0.04), health perception (bad health/not bad health) (B = −0.09; *p* = 0.01), dependency level (B = 0.08; *p* = 0.01), caregiver burden (B = 0.53; *p* = 0.00), total quality of life (B = −0.20; *p* = 0.00), and satisfaction with social support (B = −0.13; *p* = 0.00) are predictors of caregiver difficulty. These variables explained 70% of the variance of caregiver difficulty [R^2^ = 0.70; R^2^ aj = 0.69; F(12, 351) = 69.36; *p* < 0.00], with caregiver overload proving to be the strongest predictor (Table 12).

We found that the vast majority of informal caregivers have no access to or knowledge about the support provided for in the Caregiver’s Statute. The topics mentioned by more than 30% of the caregivers are related to: information on the evolution of the disease of the person being cared for (71%); information on the person being cared for by health professionals (43%); and information on the rights of the person being cared for (35%). With regard to the remaining topics, a minority of them reported having access or knowledge, namely the lack of information on referrals to appropriate services for the specific situation (6%), advice or monitoring in the area of social action (7%), participation in self-help groups (6%), access to subsidies to support the informal caregiver (6%), and whether they have been heard in the definition of public policies related to informal caregivers (5%) (Table 13).

The frequency of responses analysis of the COVID-19 consequences in relation to caregivers and the person being cared for reveals that 81% of the informal caregivers reported more difficulty in accessing consultations and 65% in accessing services and support. There is also a worsening in the caregiver’s anxiety levels (41%), as well as in their levels of worry (56%). There is an increase in the degree of isolation of the caregiver (69%) and the person being cared for (75%). Isolation and difficult access to health and other services led to increased symptomatology/needs/specificities of the individual being cared for, according to 52% of the informal caregivers. A minority report that the pandemic has limited the time/contact with the individual cared for (15%) and has reduced the existence of affection with the individual cared for (17%).

## 4. Discussion

This study aims to characterize and understand the difficulties experienced by informal caregivers in terms of relational problems (RP), social constraints (SR), care demands (CE), reactions to caregiving (RC), family support (F), and professional support. We intend to characterize these difficulties from a biopsychosocial and environmental perspective, taking into account the socio-demographic and health characteristics of the informal caregiver and the person cared for, quality of life, perceived burden, social support, and the impact of the COVID-19 pandemic on the informal caregiver and the person cared for.

We counted a group of 371 informal caregivers at the national level, mostly women and married or in a consensual union, with children. About half of them have a high level of professional activity. The vast majority are informal primary caregivers for the first time. Sociodemographic changes are found to be reflected in the number of informal caregivers [1]. In Portugal, it is estimated that about 1 million people (10% of the population) play the role of informal caregivers, which is mostly performed by women (692,305) [12].

The study integrates several variables in order to deepen knowledge about the health and difficulties of informal caregivers from a bio-psychosocial and environmental perspective.

The perception of quality of life and its dimensions were considered psychological variables, and satisfaction with social support and its dimensions were considered a social component. In this way, the analysis of the psychosocial variables under study revealed higher values in the dimensions of total, psychological, and physical quality of life. In the satisfaction with social support, family and friends emerged as the most relevant.

The difficulties experienced by caregivers have an impact on their quality of life and activities of daily living (22,28,32), with quality of life being a predictor of caregivers’ mental health [28]. When high levels of depression occur, there is a greater propensity for abandonment or lower effectiveness and satisfaction in the caregiver role to occur [19,48].

We found that the quality of life of informal caregivers is negatively correlated with the perception of caregiver burden and the perception of difficulties by the caregiver, and positively correlated with a higher perception of social support, specifically with intimacy.

In our study, we chose to assess types of social support, namely from family, friends, intimacy, and social support activities. We found that there is a negative correlation between social support and the difficulties experienced by the informal caregiver. The highest correlation is associated with intimacy, followed by friends, activities, and family. It could also be important to categorize social support into four types in terms of its functionality: emotional, tangible, informational, and companionship support in order to assess whether they differentially influence the well-being of the informal caregiver [19].

A deeper analysis of the dimensions of the difficulties experienced by the caregivers revealed that the major difficulties experienced are related to the demands of caring, difficulties at the level of family support, and professional support.

Caregiving can affect informal caregivers at psychological, professional, and social levels, impacting their health, well-being, and quality of life [1,17,18,23,24,25]. Being an informal caregiver is strongly related to decreased quality of life and well-being, high levels of depression, greater financial burden, greater impact at the physical level, reduced quality of interpersonal relationships and leisure time, and increased burnout [19,26,27,28,29,30].

The analysis of the differences according to gender and age in the difficulties in caring found significantly higher levels in women compared to men and in older people (51 years or more) compared to caregivers aged 50 years or less. Women showed higher levels in the dimensions of the perception of social constraints in their reactions to caregiving and the lack of professional support, and older people reported greater difficulties in all dimensions.

In the analysis of the caregiver’s difficulties, taking into account the professional situation, we found significant differences in the perception of relational problems and professional support, with the non-active caregivers showing higher values when compared to the non-active caregivers.

When the different levels of schooling are compared in terms of social problems and social constraints, greater difficulties are found in the group with higher education when compared to compulsory education.

The sociodemographic characteristics of informal caregivers may also function as factors related to risk and protection. The results show that being a woman, being older, not having a professional activity, and having a higher education can lead to more difficulties. Focusing on the greatest difficulties experienced by caregivers with higher levels of education, particularly related to social problems and restrictions probably resulting from a greater conciliation of the informal caregiver’s activity with the remaining family and professional obligations. In addition, in most cases, caregivers perform several services for which they have no training and receive no type of remuneration [13,14,15,16]. In addition, the role of informal caregiver is mostly played by women, middle-aged individuals, unemployed or domestic workers, and individuals with some health problems [17,18]. We found that informal caregivers with more difficulties are women, older, with more health problems, and with a lower level of education. The fact is that informal caregivers themselves often have special health and financial needs. This makes it essential to provide in-depth, effective support to informal caregivers in terms of their health, knowledge development, and skills development about the support (social, economic, health, and employment) available. In Portugal, Law No. 100/2019 contemplates all the necessary support; however, it is very difficult to access this support. Due to the lack of knowledge of informal caregivers, the lack of dissemination of support through the appropriate channels, the complexity and length of bureaucratic processes, and, in the end, the lack of resources of public services to respond.

When studying the dimensions of the difficulties perceived by the caregiver according to the perception of the caregiver’s quality of life and health, significant differences were found in all dimensions, with higher values in the group of poor quality of life and negative perception of their health.

In order to understand how the variables under study explain the difficulties experienced by informal caregivers in their different dimensions, we found that higher education, worse perception of quality of life and health, level of dependence, higher burden, and less social support explain greater difficulties experienced by the caregiver, highlighting the stronger role of the burden of the informal caregiver.

The difficulties experienced by the informal caregiver occur more often when they assess care provision as something negative and stressful [31]. This is influenced by the emotional, social, and financial stress imposed by the specificities of the person cared for, such as the duration of care, the amount of care provided per day, and the level of dependence of the individual cared for [28,29,30,31]. The level of dependence from neurodegenerative disorders, some oncological diseases, physical disabilities, and the presence of comorbidities is associated with a high caregiver burden [31].

However, if, as mentioned so far, caregiving may represent a stressful situation and entail negative consequences for the caregiver, on the other hand, it may represent benefits for the caregiver, such as the perception of a positive feeling of reward or a closer relationship with the individual being cared for [18], which may lead to an increase in life satisfaction and, consequently, a reduction in depressive feelings [19].

Social support emerges as a protective factor for both the informal caregiver and the person being cared for. Most of the time, there is one primary caregiver who may have more or less support from other secondary caregivers. An informal primary caregiver without support from other caregivers or community support is at a higher risk for his/her mental health and consequently for the caregiver’s mental health and well-being [47,48]. In this sense, the new Law No. 100/2019 for informal caregivers includes the replacement of the informal caregiver in certain situations of need and during periods of rest/holidays. It works as a protective factor in that a greater diversity and quantity of informal caregivers make the support more sustainable and comprehensive [49]. Social support plays an important role in improving quality of life, reducing the perception of burden, as well as providing better care conditions for the dependent person [21].

In Portugal in 2019, a new law, Law No. 100/2019, was published regarding the Informal Caregiver Statute. The present study aims to understand to what extent this new law is having an impact and practical applicability in supporting informal caregivers and persons cared for. We find that in 2021, only a minority are under Law No. 100/2019. We find that the vast majority of informal caregivers do not have access to or knowledge of the supports provided by the caregiver policies. The topics mentioned by more than 30% of the caregivers are related to: information about the evolution of the disease of the person being cared for (71%); information about the person being cared for by health professionals (43%); and information about the rights of the person being cared for (35%). With regard to the remaining topics, a minority reported having access or knowledge, namely the lack of information on referrals to appropriate services for the specific situation (6%), social action advice or support (7%); participation in self-help groups (6%); access to subsidies to support the informal caregiver (6%); and whether they have been heard in the definition of public policies related to informal caregivers (5%).

Returning to the research questions, in relation to “Will the informal caregiver’s sociodemographic characteristics, health, quality of life, and social support influence his/her perception of difficulties as a caregiver?” We found that, yes, the perception of difficulties is explained by the level of education, the health and quality of life of the informal caregiver, the level of dependence of the caregiver, the perception of burden, and the satisfaction with social support. Comparing the groups, we found that the difficulties are more experienced by women, caregivers who do not maintain a professional activity, older caregivers, with more health problems, and a higher educational level. Finally, in relation to the other research question, “What supports and benefits were most used by caregivers in the context of the COVID-19 pandemic?” We see that the vast majority do not make use of the rights and benefits of Law No. 100/2019. The most used resources and support in the context of the COVID-19 pandemic were information related, namely, information on the evolution of the disease of the person being cared for; information on the person being cared for by health professionals; and information on the rights of the person being cared for. On the other hand, in identifying the greatest difficulties experienced by more than half of informal caregivers in the context of the pandemic, we found that the informal caregivers reported more difficulty in accessing consultations and in accessing services and support. There is also a worsening of their levels of worry. There is an increase in the degree of isolation between the caregiver and the person being cared for. Isolation and difficult access to health and other services led to increased symptomatology/needs/specificities of the individual being cared for, according to more than half of the informal caregivers.

The results show that the COVID-19 pandemic and the respective restrictions and periods of confinement had a special and strong impact on informal caregivers and the people being cared for. The caregivers are an important part of care provision, and it is essential to support the caregivers in managing the difficulties experienced [1,21,48,50].

Informal caregivers and people being cared for were among those most affected by the pandemic and its mitigation measures. On the one hand, with their weakened health, they had to be isolated for much longer, i.e., the duration of isolation measures for the chronically ill was longer than for the rest of the population. On the other hand, being people with greater needs in terms of health and social support, they were deprived of this support in full, or even totally. Informal caregivers and people being cared for, experienced uncertainty, fear, and isolation more than other people [34,35,36,37]. The difficulties experienced by caregivers were exacerbated by the COVID-19 pandemic. The task of being an informal caregiver is in itself a physically, emotionally, and economically difficult one. With the pandemic, informal caregivers experienced even less support, less formal and informal social support, more challenges, more stress, and often helplessness without knowing how to manage the situation or how to seek help.

## 5. Conclusions

The study finds that only 15% have access to the Informal Caregiver Statute (Law No. 100/2019) and that the COVID-19 pandemic has worsened the situation of informal caregivers in terms of access to care, health status, isolation, anxiety, and worry.

The major needs felt by caregivers (mentioned by less than 20% to 10% of the participants) are associated with: benefiting from counseling, monitoring, and training for the development of care competencies, participating in the elaboration of a specific health intervention plan for the individual under their care, having a health professional as a reference contact, benefiting from rest periods, and having psychological or psychosocial support from the health/social services.

The greatest difficulties are associated with social constraints, the reactions of the person being cared for, and a lack of family support.

Higher risk groups are identified, namely, being a woman, being the main caregiver, having health problems, and having low education.

The study presents some limitations, namely, that it reveals a statistical analysis of somewhat oversimplified data; it is an exploratory study with very rich and original data; in the future, it is intended to move to more specific and deeper analysis, but the results obtained are considered an important contribution. Since we have taken into account the new law on informal caregivers, which only covers informal family caregivers, we have chosen to only consider these. This may be considered a limitation, as neighbors and the community also play an important role as informal caregivers and supporters. In a future study, we propose to expand the scope of informal caregivers to include other people who also have this function but are not included in this study. On the other hand, the present study used social support provided by family, friends, intimacy, and social support activities. The study does not cover all forms of social support; this aspect can be complemented in qualitative studies through interviews with caregivers to understand if these have different impacts on people’s psychological well-being and how. Accessing the population of informal caregivers is not easy, in a pandemic context, and with the use of an online questionnaire, the task becomes even more difficult. We believe that the most accurate way to access the population would be through associations and formal networks of caregivers. As the dissemination of the questionnaire was made by entities related to caregivers, it was not possible for us to have a response rate since we do not know how many people received the questionnaire. Another limitation related to this aspect is that we were not able to compare those who chose to answer and those who chose not to answer. We believe that the sample obtained in the adverse context in which it took place is considerable and mitigates to some extent the above-mentioned limitations.

We conclude that there is still little use of the new law on the caregiver and that there are areas that need priority intervention, namely at the level of more individual counseling, a more specific intervention plan, professional reference, and psychological and social support, all indicators of the need for greater humanization of care and services.

The lack of social support and activities related to social support, as well as the difficulty in obtaining professional support, impair the quality of life of caregivers and aggravate the perception of burden.

The COVID-19 pandemic has exacerbated the difficulties of informal caregivers, making it critical to mitigate and promote better care, better access, and a focus on promoting quality of life and the well-being of caregivers and persons cared for.

## Figures and Tables

**Table 1 ijerph-20-05205-t001:** Participants’ sociodemographic data.

	n	%	Min	Max	Average	Standard Deviation
Age	371	-	25	85	53.17	11.45
Gender						
Female	300	80.90%				
Male	71	19.10%				
Marital status						
Single	54	14.60%				
Union in fact	38	10.20%				
Married	195	52.60%				
Separated	8	2.20%				
Divorced	60	16.20%				
Widow(er)	16	4.30%				
Children						
Yes	312	84.10%				
No	59	15.90%				
Level of education						
Did not complete basic education	2	0.5%				
1ºCiclo	14	3.80%				
2nd cycle	20	5.40%				
3rd cycle	35	9.40%				
Secondary education	121	32.60%				
Degree	133	35.80%				
Master’s degree	39	10.50%				
PhD	7	1.90%				
Professional status						
Active	207	55.80%				
Unemployed	94	25.30%				
Retired	58	15.60%				
Retired with active professional Status	12	3.20%				
Region of residence						
North	78	21%				
Centre	46	12.40%				
Lisbon and Tagus Valley	191	51.50%				
Alentejo	30	8.10%				
Algarve	17	4.60%				
Autonomous Region of Madeira	3	0.80%				
Autonomous Region of the Azores	6	1.60%				
Permanent residence						
Home ownership	279	75.20%				
House for rent	66	17.80%				
Another	26	7%				

**Table 2 ijerph-20-05205-t002:** Characterization of the carer role.

	n	%
First time carer (yes)	276	74.40%
Simultaneous provision of care to more than one person (Yes)	85	22.90%
Main caregiver (yes)	302	81.40%
Gender of the family member being cared for		
Female	205	55.30%
Male	166	44.70%
Reason for dependency		
Accident	13	3.50%
Old age	58	15.60%
Disease	214	57.70%
Another	86	23.20%
Knowledge of the Caregiver’s Act (Yes)	51	13.70%

**Table 3 ijerph-20-05205-t003:** Descriptive results of the psychological variables under study.

Dimension	Min	Max	Average	Standard Deviation
Quality of life				
Total	1.08	5	3.26	0.7
Physics	1.29	5	3.4	0.76
Psychological	1	5	3.41	0.88
Social	1	5	2.98	0.95
Environmental	1	5	3.13	0.74
Satisfaction social support				
Total	1	5	2.96	0.77
Intimacy	1	5	2.86	0.89
Family	1	5	3.48	1.09
Activities	1	5	2.39	1.01
Friends	1	5	3.05	1.01
Overload	1	5	2.96	0.77
Total	21			
Loss of control	5	103	58.09	15.51
Sacrifice	6	25	14.53	4.35
Dependency	5	30	15.01	5.63
Fear/anxiety	3	25	16.39	4.02
Self-criticism	2	15	6.82	3.01
Difficulties of the carer		10	5.33	2.26
Total	1			
Relational problems	1	4	2.28	0.7
Social restrictions	1	4	1.96	0.75
Caregiving demands	1	4	2.43	0.83
Reactions to caring	1	4	2.26	0.8
Family support	1	4	2.4	0.84
Professional support	1	4	2.38	0.92
Level of dependency	0	4	2.76	0.99
		100	45.65	31.53

**Table 4 ijerph-20-05205-t004:** Results of the study of the association between the dimensions of quality of life, satisfaction with social support, difficulties of the caregiver, and level of dependence.

Variables	1	2	3	4	5	6	7	8	9	10	11	12	13
1. QL—Total	-												
2. QL—Physics	0.88 **	-											
3. QL—Psychological	0.90 **	0.72 **	1										
4. QL—Social	0.79 **	0.57 **	0.70 **	-									
5. QL—Environmental	0.89 **	0.70 **	0.68 **	0.64 **	1								
6. Overload	−0.52 **	−0.46 **	−0.46 **	−0.54 **	−0.42 **	-							
7. SSS—Total	0.58 **	0.44 **	0.50 **	0.64 **	0.51 **	−0.57 **	-						
8. SSS—Intimacy	0.51 **	0.36 **	0.43 **	0.54 **	0.50 **	−0.44 **	0.80 **	-					
9. SSS—Family	0.34 **	0.30 **	0.29 **	0.39 **	0.26 **	−0.39 **	0.71 **	0.37 **	-				
10. SSS—Social activities	0.45 **	0.36 **	0.40 **	0.42 **	0.41 **	−0.43 **	0.62 **	0.45 **	0.22 **	-			
11. SSS—Friends	0.47 **	0.34 **	0.39 **	0.56 **	0.40 **	−0.48 **	0.89 **	0.61 **	0.59 **	0.37 **	-		
12. Difficulties of the carer	−0.64 **	−0.57 **	−0.56 **	−0.60 **	−0.54 **	0.76 **	−0.60 **	−0.52 **	−0.40 **	−0.47 **	−0.47 **	-	
13. Level of dependency	−0.29 **	−0.30 **	−0.21 **	−0.20 **	−0.27 **	0.16 **	−0.23 **	−0.18 **	−0.10 *	−0.25 **	−0.18 **	0.31 **	-

Note: ** *p* < 0.05. * *p* < 0.01. QL—quality of life; SSS—social support satisfaction.

**Table 5 ijerph-20-05205-t005:** Study of the association between the dimensions of caregiver difficulties.

Variables	1	2	3	4	5	6	7
1. Difficulties of the carer	-						
2. Relational problems	0.83 **	-					
3. Social restrictions	0.91 **	0.63 **	-				
4. Caregiving demands	0.92 **	0.69 **	0.82 **	-			
5. Reactions to caring	0.86 **	0.61 **	0.76 **	0.77 **	-		
6. Family Support	0.79 **	0.57 **	0.81 **	0.68 **	0.63 **	-	
7. Professional support	0.60 **	0.38 **	0.51 **	0.50 **	0.52 **	0.45 **	-

Note: ** *p* < 0.01.

**Table 6 ijerph-20-05205-t006:** Results of the study of caregiver difficulties by gender.

	Female		Male			
	M	SD	M	SD	*T*	*p*
Difficulties of the carer						
Relational problems	1.97	0.76	1.92	0.70	0.44	0.66
Social restrictions	2.49	0.85	2.18	0.71	3.13	0.00
Caregiving demands	2.28	0.81	2.17	0.77	1.04	0.30
Reactions to caring	2.45	0.86	2.18	0.77	2.42	0.02
Family support	2.41	0.94	2.27	0.82	1.15	0.25
Professional support	2.85	0.98	2.40	0.96	3.46	0.00
Total	2.31	0.71	2.13	0.63	1.98	0.05

**Table 7 ijerph-20-05205-t007:** Results of the study of caregiver difficulties as a function of professional activity (active/not active).

	Active Profession (n = 207)		Non Active Profession (n = 164)			
	M	SD	M	SD	T	*p*
Difficulties of the carer						
Relational problems	1.65	0.74	1.96	0.75	−0.15	0.00
Social restrictions	2.43	0.84	2.43	0.82	0.88	0.99
Caregiving demands	2.22	0.82	2.29	0.77	−0.81	0.42
Reactions to caring	2.41	0.84	2.34	0.85	0.34	0.74
Family support	2.37	0.94	2.37	0.94	−0.19	0.85
Professional support	2.64	0.99	2.92	0.97	−2.79	0.01
Total	2.26	0.71	2.30	0.68	−0.46	0.65

**Table 8 ijerph-20-05205-t008:** Results of the study of caregiver difficulties as a function of education.

	Primary School (n = 192)		Tertiary School (n = 172)			
	M	SD	M	SD	T	*p*
Difficulties of the carer						
Relational problems	1.86	0.73	2.04	0.73	−2.45	0.02
Social restrictions	2.31	0.81	2.54	0.82	−2.66	0.01
Caregiving demands	2.18	0.77	2.33	0.81	−1.80	0.07
Reactions to caring	2.38	0.84	2.40	0.84	−0.24	0.81
Family support	2.32	0.89	2.44	0.93	−1.24	0.22
Professional support	2.79	0.97	2.72	1.01	0.74	0.46
Total	2.21	0.69	2.34	0.675	−1.92	0.06

**Table 9 ijerph-20-05205-t009:** Results of the study of the difficulties of the caregiver as a function of the perception of quality of life.

	Poor Quality of Life (n = 91)		Not Poor Quality of Life (n = 280)			
	M	SD	M	SD	T	*p*
Difficulties of the carer						
Relational problems	2.40	0.79	1.82	0.67	6.29	0.00
Social restrictions	3.18	0.63	2.18	0.736	11.63	0.00
Caregiving demands	2.95	0.63	2.03	0.72	10.92	0.00
Reactions to caring	3.08	0.61	2.17	0.79	11.52	0.00
Family support	3.05	0.75	2.16	0.85	9.50	0.00
Professional support	3.38	0.65	2.56	1.00	9.03	0.00
Total	2.91	0.54	2.08	0.61	11.56	0.00

**Table 10 ijerph-20-05205-t010:** Results of the study of caregiver difficulties as a function of health perception.

	Poor Health (n = 131)		Not Poor Health (n = 240)			
	M	SD	M	SD	T	*p*
Difficulties of the carer						
Relational problems	2.23	0.79	1.81	0.68	5.11	0.00
Social restrictions	2.88	0.80	2.18	0.74	8.51	0.00
Caregiving demands	2.75	0.73	1.98	0.70	9.94	0.00
Reactions to caring	2.90	0.74	2.12	0.77	9.36	0.00
Family support	2.79	0.92	2.15	0.83	6.82	0.00
Professional support	3.19	0.83	2.53	0.99	6.87	0.00
Total	2.70	0.65	2.05	0.61	9.43	0.00

**Table 11 ijerph-20-05205-t011:** Results of the study of caregiver difficulties as a function of age of participants.

	Up to 50 Years (n = 156)		51 and over (n = 215)			
	M	SD	M	SD	T	*p*
Difficulties of the carer						
Relational problems	1.91	0.74	2.00	0.75	−1.11	0.27
Social restrictions	2.40	0.82	2.45	0.84	−0.56	0.57
Caregiving demands	2.25	0.79	2.26	0.81	−0.04	0.97
Reactions to caring	2.52	0.85	2.31	0.83	2.37	0.02
Family support	2.36	0.89	2.39	0.94	−0.32	0.75
Professional support	2.76	1.02	2.76	0.96	−0.08	0.93
Total	2.28	0.69	2.28	0.70	−0.02	0.98

**Table 12 ijerph-20-05205-t012:** Studying gender, age, marital status, education, employment status, quality of life, knowledge of caregiver law, level of dependency, and perceived burden as predictors of caregiver difficulties.

Variables	R^2^	R^2^ Adjusted	Non-Standardized Coefficients		Standardized Coefficients	t	*p*
			Beta	Standard Error	Beta		
(Constant)	0.70	0.69	1.94	0.28		7.03	0.00
Gender			0.03	0.05	0.02	0.57	0.57
Age			−0.00	0.00	-0.05	−1.50	0.14
Marital status			0.05	0.04	0.03	1.13	0.26
Education			0.084	0.04	0.06	2.02	0.04
Professional status			0.012	0.05	0.01	0.25	0.80
Perception of health (bad health/not bad health)			−0.13	0.05	−0.09	−2.47	0.01
Knowledge Caregiver Act			0.04	0.06	0.01	0.59	0.55
Level of dependency			0.00	0.00	0.08	2.70	0.01
Caregiver overload			0.47	0.03	0.53	13.78	0.00
Quality of life			−0.19	0.04	−0.20	−4.55	0.00
Satisfaction social support			−0.12	0.04	−0.13	−3.43	0.00

**Table 13 ijerph-20-05205-t013:** Analysis of the frequency of answers to information on informal caregiver status.

	n	%
Are you under Law No. 100/2019 on the Statute of the Informal Caregiver? (Yes)	51	13.7
Do you usually receive information concerning the individual under your care from health professionals? (Yes)	160	43.1
Do you have access to information that enlightens you about the evolution of the illness of the dependent individual? (Yes)	263	70.9
Have you ever been assigned a health professional as a reference contact? (Yes)	69	18.6
Do you benefit or have you benefited from counselling, coaching and training for the development of caring skills by health professionals? (Yes)	61	16.4
Do you participate or have you ever participated in the development of a specific intervention plan in the area of health, directed at the individual under your care? (Yes)	65	17.5
Do you usually receive information about the individual you care for from social workers? (Yes)	32	8.6
Do you have access to information that will tell you about all the support the cared-for individual is entitled to? (Yes)	129	34.8
Do you benefit or have you ever benefited from advice and guidance about your rights and responsibilities as an informal caregiver, and the rights and responsibilities of the individual being cared for, from the competent services of SS? (Yes)	29	7.8
Have you ever received information or been referred by the competent social security services to other services appropriate to your particular situation (only in cases where it is justified)? (Yes)	23	6.2
Have you ever benefited from counselling and accompaniment from professionals in the area of social security or from municipalities, within the scope of direct social action services? (Yes)	25	6.7
Since you started your job as a caregiver, have you ever received psychosocial support? (Yes)	36	9.7
Since you started your job as a caregiver, have you ever received psychological support from the health services? (Yes)	44	11.9
Since you started your role as a caregiver have you participated in any self-help groups developed by the health services? (Yes)	21	5.7
Since starting care, have you ever received information or been referred to social support networks (such as home help)? (Yes)	56	15.1
Since you began caring, have you ever taken one or more rest periods? (Yes)	72	19.4
Have you ever received informal caregiver support allowance? (Yes)	21	5.7
Do you feel you can reconcile caring with your professional life? Yes Not applicable	163 70	43.9 18.9
Do you feel that your role in maintaining the well-being of the individual cared for is properly recognized? (Yes)	81	21.8
Have you ever been heard in the development of public policy for informal caregivers? (Yes)	17	4.6

## Data Availability

Data is unavailable due to privacy or ethical restrictions.

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
