# Peer review of "Relationship between Burden, Quality of Life and Difficulties of Informal Primary Caregivers in the Context of the COVID-19 Pandemic: Analysis of the Contributions of Public Policies"

_ijerph, 2023, doi:10.3390/ijerph20065205_

Round 1

Reviewer 1 Report

Thanks for inviting me to review this manuscript. This is an important topic and I think it is a well written paper in general. My detailed comments:

1.        In the introduction section, the authors did not mention Covid-19 until the last sentence of the section. The authors should clearly state the very important role informal caregivers have played during the pandemic (I think it’s better to be after line 55). As Liu et al. (2021) revealed informal caregivers were extremely important in the early phase of the pandemic when people’s mobility and everyday activities have been considerably influenced by lockdown measures and other containment interventions. Vulnerable groups (such as older people and low income people) were highly dependent on informal caregivers. And Informal caregivers have contributed to Covid containment and recovery. That’s why this paper would focus on them.

2.        My major concern is that a large part of the results section presents very descriptive results (correlation analyses and ANOVA tests, also, are usually not deemed as the results of a study). The only one that could be seen as a proper analysis is a very simple linear regression analysis. I am not saying that complex methods are always better than simpler ones but I think it’s a bit oversimplified. For example, if you want to compare the two groups (Table 6-11), you could at least try propensity score matching, otherwise the results of this study are vulnerable to self-selection bias. I am aware that it may be difficult to do the analyses all over again, so another way to deal with this could be to add this as one of the limitations of this study.

3.        In this paper, the authors considered some particular types of social support, but there are other kinds of support such as support from the neighbourhood and government-provided social support. Previous studies have confirmed that these forms of social support helped people to cope with Covid-related psychological problems (e.g., Chen et al., 2021; Jones et al., 2020). Why did you exclude these forms of social support?

4.        Social support is also commonly decomposed into four types in terms of its functionality: emotional, tangible, informational, and companionship support (Uchino, 2004). Would these have different impacts on people’s psychological wellbeing, and how? I am aware that you may not have sufficient data for such analyses but please at least list this as one of the limitations.

5.        People may have received different forms of informal care in very different ways. For example, Liu et al. (2022a) indicate that rural-urban migrants in China relied on family members during Covid but they can hardly acquire any care from friends, community etc. Would receiving informal care from one single source be good enough for maintaining a good mental health status? Or disadvantaged people need various sources of informal care? Discuss this in section 4.

Reference

Chen, X., Zou, Y., & Gao, H. (2021). Role of neighborhood social support in stress coping and psychological wellbeing during the COVID-19 pandemic: Evidence from Hubei, China. Health & Place, 69, 102532.

Jones, M., Beardmore, A., Biddle, M., Gibson, A., Ismail, S. U., McClean, S., & White, J. (2020). Apart but not Alone? A cross-sectional study of neighbour support in a major UK urban area during the COVID-19 lockdown. Emerald Open Research, 2.

Uchino, B. N. (2004). Social support and physical health: Understanding the health consequences of relationships. Yale university press.

Author Response

Revisions Paper

“Relationship between burden, quality of life and difficulties of informal primary caregivers in the context of the pandemic COVID-19: analysis of the contributions of public policies”

Dear Editor and Reviewer

I greatly appreciate the careful analysis and recommendations indicated by the reviewers. I have made the requested changes; in the response I indicate in yellow the changed text and in the manuscript, I indicate in track changes. I made substantial changes that greatly improved the paper, thank you very much. If you have any further questions or if needed any additional changes, please just let me know best regards Tania

Reviewer 1

  1. In the introduction section, the authors did not mention Covid-19 until the last sentence of the section. The authors should clearly state the very important role informal caregivers have played during the pandemic (I think it’s better to be after line 55). As Liu et al. (2021) revealed informal caregivers were extremely important in the early phase of the pandemic when people’s mobility and everyday activities have been considerably influenced by lockdown measures and other containment interventions. Vulnerable groups (such as older people and low-income people) were highly dependent on informal caregivers. And Informal caregivers have contributed to Covid containment and recovery. That’s why this paper would focus on them.

The COVID-19 pandemic, the respective isolation measures, difficulties in accessing health and social services that were focused exclusively on fighting the pandemic at the time, had a greater impact on certain groups, namely the elderly, people with chronic illnesses, people with socio-economic difficulties, women and the unemployed. Informal caregivers and their cared-for persons were one of the groups most affected by the pandemic and its restrictions [34, 35, 36].

The informal caregivers were extremely important for stress management and psychological wellbeing promotion during the pandemic when people’s mobility and everyday activities have been considerably influenced by lockdown measures and other containment interventions. Vulnerable groups (such as older people and low-income people) were highly dependent on informal caregivers. And Informal caregivers have contributed to Covid containment and recovery [37].

  1. My major concern is that a large part of the results section presents very descriptive results (correlation analyses and ANOVA tests, also, are usually not deemed as the results of a study). The only one that could be seen as a proper analysis is a very simple linear regression analysis. I am not saying that complex methods are always better than simpler ones but I think it’s a bit oversimplified. For example, if you want to compare the two groups (Table 6-11), you could at least try propensity score matching, otherwise the results of this study are vulnerable to self-selection bias. I am aware that it may be difficult to do the analyses all over again, so another way to deal with this could be to add this as one of the limitations of this study.

The study reveals statistical analysis of somewhat oversimplified data, it is an exploratory study, with very rich and original data, in the future it is intended to move to more specific and deeper analysis but it is considered an important contribution the results obtained.

The study presents some limitations, namely, reveals statistical analysis of some-what oversimplified data, it is an exploratory study, with very rich and original data, in the future it is intended to move to more specific and deeper analysis but it is considered an important contribution the results obtained. Since we have taken into account the new law on informal caregivers and this only covers informal family care-givers, we have chosen to only consider these. This may be considered a limitation as neighbours and the community also play an important role as informal caregiver and support. In a future study, we propose to expand the scope of informal caregivers to include other people who also have this function and are not included in this study. On the other hand, the present study used social support provided by family, friends, intimacy and social support activities. The study does not cover all forms of social support, this aspect can be complemented in qualitative studies through interviews with caregivers to understand if these have different impacts on people's psychological wellbeing, and how. Accessing the population of informal caregivers is not easy, in a pandemic context and with the use of an online questionnaire the task becomes even more difficult. We believe that the most accurate way to access the population would be through associations and formal networks of caregivers. As the dissemination of the questionnaire was made by entities related to caregivers, it was not possible for us to have a response rate since we do not know how many people received the questionnaire. Another limitation related to this aspect is that we were not able to compare those who chose to answer and those who chose not to answer. We believe that the sample obtained in the adverse context in which it took place is considerable and mitigates to some extent the above mentioned limitations.

  1. In this paper, the authors considered some particular types of social support, but there are other kinds of support such as support from the neighbourhood and government-provided social support. Previous studies have confirmed that these forms of social support helped people to cope with Covid-related psychological problems (e.g., Chen et al., 2021; Jones et al., 2020). Why did you exclude these forms of social support?

Since we have taken into account the new law on informal caregivers and this only covers informal family caregivers, we have chosen to only consider these. This may be considered a limitation as neighbours and the community also play an important role as informal caregiver and support. In a future study, we propose to expand the scope of informal caregivers to include other people who also have this function and are not included in this study.

4.Social support is also commonly decomposed into four types in terms of its functionality: emotional, tangible, informational, and companionship support (Uchino, 2004).

Would these have different impacts on people’s psychological wellbeing, and how? I am aware that you may not have sufficient data for such analyses but please at least list this as one of the limitations.

In our study, we chose to assess types of social support, namely from family, friends, intimacy and social support activities. We found that there is a negative correlation between social support and the difficulties experienced by the informal caregiver. The highest correlation is associated with intimacy, followed by friends, activities and family. It could also be important to assess social support into four types in terms of its functionality: emotional, tangible, informational, and companionship support (Uchino, 2004) in order to assess whether they differentially influence the well-being of the informal caregiver.

On the other hand, the present study used social support provided by family, friends, intimacy and social support activities. The study does not cover all forms of social support, this aspect can be complemented in qualitative studies through interviews with caregivers to understand if these have different impacts on people's psychological wellbeing, and how?

  1. People may have received different forms of informal care in very different ways. For example, Liu et al. (2022a) indicate that rural-urban migrants in China relied on family members during Covid but they can hardly acquire any care from friends, community etc. Would receiving informal care from one single source be good enough for maintaining a good mental health status? Or disadvantaged people need various sources of informal care? Discuss this in section 4.

Social support emerges as a protective factor for both the informal caregiver and the person being cared for. Most of the time there is one primary caregiver who may have more or less support from other secondary caregivers. An informal primary caregiver without support from other caregivers or community support is at a higher risk for his/her mental health and consequently for the caregiver's mental health and well-being. In this sense the new law for informal caregiver includes replacement of the informal caregiver in certain situations of need and in periods of rest/holidays. It works as a protective factor that a greater diversity and quantity of informal caregivers makes the support more sustainable and comprehensive.

Bergmann M and Wagner M (2021). The Impact of COVID-19 on Informal Caregiving and Care Receiving Across Europe During the First Phase of the Pandemic. Frontiers in Public Health 9:673874. doi: 10.3389/fpubh.2021.673874

Reference

Chen, X., Zou, Y., & Gao, H. (2021). Role of neighborhood social support in stress coping and psychological wellbeing during the COVID-19 pandemic: Evidence from Hubei, China. Health & Place, 69, 102532. doi: 10.1016/j.healthplace.2021.102532.

Jones, M., Beardmore, A., Biddle, M., Gibson, A., Ismail, S. U., McClean, S., & White, J. (2020). Apart but not Alone? A cross-sectional study of neighbour support in a major UK urban area during the COVID-19 lockdown. Emerald Open Research, 2.

Uchino, B. N. (2004). Social support and physical health: Understanding the health consequences of relationships. Yale university press.

Reviewer 2 Report

Dear authors. First, I congratulate you on the theme, which is relevant because of the aging population and the high demand for family caregivers, who are often made invisible. However, I send some suggestions to improve the text.

Title - does not reflect the text.

Abstract: The objective here differs from that presented in the introduction. 

Introduction - problematize the context of covid 19 for these caregivers. Swap the order of objectives: first characterize, then understand. Consider bringing up the research question.

Method - Explain better how the data collection took place, and make it clear why you used the Barthel scale and with whom? Put tables 1 and 2 in the results part.

Results - To bring in these the relation of ages, considering the profile of middle-aged and older women with their demands

Discussion: You could better compare your findings with those of other studies. It did not bring the context of COVID-19. The objective is to understand and characterize how the raised demands reflect on caregivers.

So look at the demands of the middle-aged and elderly (their context), certainly in more significant numbers.

Final considerations - return to the research question to answer it. 

References are primarily current; however, I suggest reducing, using the main ones, and better discussing these with your results.

Author Response

Revisions Paper

“Relationship between burden, quality of life and difficulties of informal primary caregivers in the context of the pandemic COVID-19: analysis of the contributions of public policies”

Dear Editor and Reviewer

I greatly appreciate the careful analysis and recommendations indicated by the reviewers. I have made the requested changes; in the response I indicate in yellow the changed text and in the manuscript, I indicate in track changes. I made substantial changes that greatly improved the paper, thank you very much. If you have any further questions or if needed any additional changes, please just let me know best regards Tania

Reviewer 2

Title - does not reflect the text.

“Difficulties of informal primary caregivers in the context of the pandemic COVID-19: a bio-psychosocial and environmental approach”

Abstract: The objective here differs from that presented in the introduction. 

The study aimed to understand and characterize the difficulties experienced by informal caregivers from a bio-psychosocial and environmental perspective, taking into account the socio-demographic and health characteristics of the informal caregiver and the person cared for, quality of life, perceived burden, social support, and the impact of the Covid-19 Pandemic on the informal caregiver and the person cared for.

Introduction - problematize the context of covid 19 for these caregivers. Swap the order of objectives: first characterize, then understand. Consider bringing up the research question.

The COVID-19 pandemic, the respective isolation measures, difficulties in accessing health and social services that were focused exclusively on fighting the pandemic at the time, had a greater impact on certain groups, namely the elderly, people with chronic illnesses, people with socio-economic difficulties, women and the unemployed. Informal caregivers and their cared-for persons were one of the groups most affected by the pandemic and its restrictions [34, 35, 36].

As Liu et al. [37] revealed informal caregivers were extremely important in the early phase of the pandemic when people’s mobility and everyday activities have been con-siderably influenced by lockdown measures and other containment interventions. Vulnerable groups (such as older people and low-income people) were highly dependent on informal caregivers. And Informal caregivers have contributed to Covid containment and recovery.

The main objective is to characterise and understand the difficulties experienced by informal caregivers in the context of Pandemic COVID-19 from a bio-psychosocial and environmental perspective.  The difficulties experienced by informal caregivers was considered in terms of relational problems (RP), social constraints (SR), care demands (CE), reactions to caregiving (RC), family support (F) and professional support. We aimed to characterize these difficulties from a bio-psychosocial and environmental perspective, taking into account the socio-demographic and health characteristics of the informal caregiver and the person cared for, quality of life, perceived burden, social support, and the impact of the Covid-19 Pandemic on the informal caregiver and the person cared for. The following research questions can be asked: Will the informal caregiver's socio-demographic characteristics, health, quality of life, and social support influence his/her perception of difficulties as a caregiver? What supports and benefits were most used by caregivers in the context of pandemic COVID-19?

Method - Explain better how the data collection took place, and make it clear why you used the Barthel scale and with whom?

The data collection procedure consisted first of contacting several organizations and associations related to informal caregivers, of convenience such as CERCI's, associations supporting informal caregivers, associations supporting specific groups (e.g. cerebral palsy, autism, among others) and support groups, to support the research team in disseminating the questionnaire to informal caregivers. The study was developed during the context of the COVID-19 pandemic (march 2021 to December 2021) which made it very difficult to access the study population.

Barthel Index was used to assess the level of dependence of the person cared for. It was completed by the informal caregiver, reporting the person cared for characteristics

Put tables 1 and 2 in the results part.

changed

Results - To bring in these the relation of ages, considering the profile of middle-aged and older women with their demands

We found that informal caregivers with more difficulties are women, older, with more health problems and with a lower level of education. The fact is that informal caregivers themselves often have special health and financial needs. This makes it essential to provide in-depth, effective support to informal caregivers in terms of their health, knowledge development and skills development about the support (social, economic, health and employment) available. In Portugal, the Law No. 100/2019 contemplates all the necessary support, however, it is very difficult to access this support. Due to the lack of knowledge of informal caregivers, the lack of dissemination of support through the appropriate channels, the complexity and length of bureaucratic processes and, in the end, the lack of resources of public services to respond.

Discussion: You could better compare your findings with those of other studies. It did not bring the context of COVID-19. The objective is to understand and characterize how the raised demands reflect on caregivers. So look at the demands of the middle-aged and elderly (their context), certainly in more significant numbers.

This study aims to characterize and understand the difficulties experienced by in-formal caregivers in terms of relational problems (RP), social constraints (SR), care de-mands (CE), reactions to caregiving (RC), family support (F) and professional support. We intend to characterize these difficulties from a biopsychosocial and environmental per-spective, taking into account the socio-demographic and health characteristics of the in-formal caregiver and the person cared for, quality of life, perceived burden, social support, and the impact of the Covid-19 Pandemic on the informal caregiver and the person cared for.

We counted a group of 371 informal caregivers at national level, mostly women and married or in a consensual union, with children. About half of them have a high pro-fessional activity. The vast majority are informal primary caregivers and for the first time. Sociodemographic changes are found to be reflected in the number of informal caregivers [1]. In Portugal, it is estimated that about 1 million people (10% of the population) play the role of informal caregivers, being mostly performed by women (692 305) [12].

The study integrates several variables in order to deepen the knowledge about the health and difficulties of informal caregivers from a bio-psychosocial and environmental perspective.

Perception of quality of life and its dimensions were considered as psychological variables and satisfaction with social support and its dimensions as social component. In this way, the analysis of the psychosocial variables under study revealed higher values in the dimensions of total, psychological and physical quality of life. In the satisfaction with social support, family and friends emerged as the most relevant.

The difficulties experienced by caregivers have an impact on their quality of life and activities of daily living (22,28,32), being quality of life a predictor of caregivers' mental health [28]. When high levels of depression occur, there is a greater propensity for abandonment or lower effectiveness and satisfaction in the caregiver role to occur [19].

We found that the quality of life of informal caregivers is negatively correlated with the perception of caregiver burden, the perception of difficulties by the caregiver and positively correlated with a higher perception of social support, specifically with intimacy.

In our study, we chose to assess types of social support, namely from family, friends, intimacy and social support activities. We found that there is a negative correlation between social support and the difficulties experienced by the informal caregiver. The highest correlation is associated with intimacy, followed by friends, activities and family. It could also be important to assess social support into four types in terms of its func-tionality: emotional, tangible, informational, and companionship support (Uchino, 2004) in order to assess whether they differentially influence the well-being of the informal caregiver.

A deeper analysis of the dimensions of the difficulties experienced by the caregivers, we found that the major difficulties experienced are related to the demands of caring, difficulties at the level of family support and professional support.

Caregiving can affect informal caregivers at psychological, professional and social levels, impacting their health, well-being and quality of life [1,17,18,23-25]. Being an informal caregiver is strongly related to decreased quality of life and well-being, high levels of depression, greater financial burden, greater impact at the physical level, re-duced quality of interpersonal relationships and leisure time, and increased burnout [19,26-30].

The analysis of the differences according to gender and age in the difficulties in caring found significantly higher levels in women compared to men, and in older people (51 years or more) compared to caregivers aged 50 years or less. Women showed higher levels in the dimensions of the perception of social constraints in the reactions to care-giving and lack of professional support, and older people reported greater difficulties in all dimensions.

In the analysis of the caregiver's difficulties, taking into account the professional situation, we found significant differences in the perception of relational problems and professional support, with the non-active caregivers showing higher values when compared to the non-active caregivers.

When the different levels of schooling are compared in the dimension’s social problems and social constraints, greater difficulties are found in the group with higher education when compared to compulsory education.

The sociodemographic characteristics of informal caregivers may also function as factors related to risk and protection. The results show that being a woman, being older, not having a professional activity and having higher education can lead to more difficulties. Focusing on the greatest difficulties experienced by caregivers with higher levels of education, particularly related to social problems and restrictions probably resulting from a greater conciliation of the informal caregiver's activity with the remaining family and professional obligations. In addition, in most cases, caregivers perform several ser-vices for which they have no training and receive no type of remuneration [13-16]. In addition, the role of informal caregiver is mostly played by women, middle-aged individuals, unemployed or domestic workers, and individuals with some health problem [17,18]. We found that informal caregivers with more difficulties are women, older, with more health problems and with a lower level of education. The fact is that informal caregivers themselves often have special health and financial needs. This makes it essential to provide in-depth, effective support to informal caregivers in terms of their health, knowledge development and skills development about the support (social, economic, health and employment) available. In Portugal, the Law No. 100/2019 con-templates all the necessary support, however, it is very difficult to access this support. Due to the lack of knowledge of informal caregivers, the lack of dissemination of support through the appropriate channels, the complexity and length of bureaucratic processes and, in the end, the lack of resources of public services to respond.

When studying the dimensions of the difficulties perceived by the caregiver ac-cording to the perception of the caregiver's quality of life and health, significant differ-ences were found in all dimensions, with higher values in the group of poor quality of life and negative perception of their health.

In order to understand how the variables under study explain the difficulties ex-perienced by informal caregivers in their different dimensions, we found that higher education, worse perception of quality of life and health, level of dependence, higher burden and less social support explain greater difficulties experienced by the caregiver, highlighting the stronger role of the burden of the informal caregiver.

The difficulties experienced by the informal caregiver occur more often when they assess care provision as something negative and stressful [31]. This is influenced by the emotional, social and financial stress imposed by the specificities of the person cared for, such as, the duration of care, the amount of care provided per day and the level of de-pendence of the individual cared for [28-31]. The level of dependence from neuro-degenerative disorders, some oncological diseases, physical disabilities and the presence of comorbidities are associated with high caregiver burden [31].

However, if, as mentioned so far, caregiving may represent a stressful situation and entail negative consequences for the caregiver, on the other hand, it may represent benefits for the caregiver, such as the perception of a positive feeling of reward or a closer relationship with the individual being cared for [18], which may lead to an increase in life satisfaction and, consequently, a reduction of depressive feelings [19].

Social support emerges as a protective factor for both the informal caregiver and the person being cared for. Most of the time there is one primary caregiver who may have more or less support from other secondary caregivers. An informal primary caregiver without support from other caregivers or community support is at a higher risk for his/her mental health and consequently for the caregiver's mental health and well-being [47]. In this sense the new law No. 100/2019 for informal caregiver includes replacement of the informal caregiver in certain situations of need and in periods of rest/holidays. It works as a protective factor that a greater diversity and quantity of informal caregivers makes the support more sustainable and comprehensive [Bergmann M and Wagner M (2021)]. Social support plays an important role in improving quality of life, reducing the perception of burden, as well as providing better care conditions for the dependent person [21].

In Portugal in 2019 a new law, Law No. 100/2019, was published regarding the Informal Caregiver Statute. The present study aims to understand to what extent this new law is having an impact and practical applicability in supporting informal caregivers and persons cared for. We find that in 2021 only a minority are under Law No. 100/2019. We find that the vast majority of informal caregivers do not have access to or knowledge of the supports provided by the caregivers policies. The topics mentioned by more than 30% of the caregivers are related to: information about the evolution of the disease the person being cared for (71%); information about the person being cared for by health profes-sionals (43%) and information about the rights of the person being cared for (35%). With regard to the remaining topics, a minority reported having access or knowledge, namely the lack of information on referrals to appropriate services for the specific situation (6%), social action advice or support (7%); participation in self-help groups (6%); access to subsidies to support the informal caregiver (6%) and whether they have been heard in the definition of public policies related to informal caregivers (5%).

Returning to the research questions, in relation to “Will the informal caregiver's sociodemographic characteristics, health, quality of life, and social support influence his/her perception of difficulties as a caregiver?” We found that, yes, the perception of difficulties is explained by the level of education, the health and quality of life of the informal caregiver, the level of dependence of the caregiver, the perception of burden and the satisfaction with social support. Comparing groups, we found that the difficulties are more experienced by women, caregivers who do not maintain a professional activity, older caregivers, with more health problems and higher educational level. Finally in relation to other research question “What supports and benefits were most used by caregivers in the context of pandemic COVID-19?” we see that the vast majority do not make use of the rights and benefits of Law no. 100/2019. The most used resources and support in the context of a pandemic COVID-19 were information related, namely, in-formation on the evolution of the disease of the person being cared for; information on the person being cared for by health professionals and information on the rights of the person being cared for. On the other hand, in identifying the greatest difficulties experienced by more than half of informal caregivers in the context of the pandemic, we found that the informal caregivers reported more difficulty in accessing consultations and in accessing services and support.  There is also a worsening their levels of worry. There is an increase in the degree of isolation of the caregiver and the person being cared for. Isolation and difficult access to health and other services led to increased symptomatolo-gy/needs/specificities of the individual being cared for according to more than half of the informal caregivers.

The results show that the Covid-19 Pandemic and the respective restrictions and periods of confinement had a special and strong impact on informal carers and people being cared for. The caregivers is an important part of care provision, and it is essential to support caregivers in managing the difficulties experienced [1,21].

Informal caregivers and people being cared were among those most affected by the pandemic and its mitigation measures. On the one hand, with their weakened health they had to be isolated for much longer, i.e. the duration of isolation measures for the chronically ill was longer than for the rest of the population. On the other hand, being people with greater needs in terms of health and social support, they were deprived of this support in full, or even totally. Informal caregivers and people being cared for, expe-rienced uncertainty, fear and isolation more than other people [34, 35, 36, 37]. The dif-ficulties experienced by caregivers were exacerbated by the COVID-19 pandemic. The task of being an informal caregiver is in itself a physically, emotionally and economically difficult one. With the pandemic, informal caregivers experienced even less support, less formal and informal social support, more challenges, more stress and often helplessness without knowing how to manage the situation or how to seek help.

Final considerations - return to the research question to answer it. 

Returning to the research questions, in relation to “Will the informal caregiver's sociodemographic characteristics, health, quality of life, and social support influence his/her perception of difficulties as a caregiver?” We found that, yes, the perception of difficulties is explained by the level of education, the health and quality of life of the informal caregiver, the level of dependence of the caregiver, the perception of burden and the satisfaction with social support. Comparing groups, we found that the difficulties are more experienced by women, caregivers who do not maintain a professional activity, older caregivers, with more health problems and lower educational level. Finally in relation to other research question “What supports and benefits were most used by caregivers in the context of pandemic COVID-19?” we see that the vast majority do not make use of the rights and benefits of Law no. 100/2019. The most used resources and support in the context of a pandemic COVID-19 were information related, namely, in-formation on the evolution of the disease of the person being cared for; information on the person being cared for by health professionals and information on the rights of the person being cared for. On the other hand, in identifying the greatest difficulties experienced by more than half of informal caregivers in the context of the pandemic, we found that the informal caregivers reported more difficulty in accessing consultations and in accessing services and support.  There is also a worsening their levels of worry. There is an increase in the degree of isolation of the caregiver and the person being cared for. Isolation and difficult access to health and other services led to increased symptomatology/needs/specificities of the individual being cared for according to more than half of the informal caregivers.

References are primarily current; however, I suggest reducing, using the main ones, and better discussing these with your results.

Reviewer 3 Report

Review

This is a very interesting study about an important issue that is often overlooked – the primary caregiver and his burden.

A few suggestions:

ABSTRACT

Please elaborate on the methods (a questionnaire study, sampling method).  

INTRODUCTION

Lines 31-32 – non-communicable diseases and chronic diseases are the same.

Lines 38-39 – this is not a clear sentence.

Lines 40-41 – please elaborate at least once on who is the informal caregiver (a definition will be good).

Please add a short explanation in the INTRO about Law No. 100/2019. You refer to it in the manuscript, but it is not clear what exactly it means.

At the end of the INTRO, the study's aim is not clearly defined. Please make shorter sentences and try to make it clear and concise.

METHODS

2.1 participants

Please mention here the sampling methods (convenient).

2.3 procedure

Please elaborate on the statistical analysis. It is lacking.

RESULTS

Please mention what was the time frame in which you collected the questionnaires.

Lines 256-260 – in the discussion, I would add an explanation to that point. Perhaps caregivers with less burden have a higher quality of life? The direction of the association should be discussed. (please add to the discussion, lines 347-349).

Discussion

Lines 340-342 – not clear.

Lines 372-376 – this is a very important sentence. But it is too long. Please rephrase and maybe make 2 sentences to make it easier for the reader to understand.

You need to add to the discussion a paragraph with the subheading: Strengths and limitations. Please add there that you do not have a response rate (you did not mention how many people received the questionnaire) and did not compare those who chose to answer and those who chose not to answer. This is a strong selection bias and should be addressed.  

Author Response

Revisions Paper

“Relationship between burden, quality of life and difficulties of informal primary caregivers in the context of the pandemic COVID-19: analysis of the contributions of public policies”

Dear Editor and Reviewer

I greatly appreciate the careful analysis and recommendations indicated by the reviewers. I have made the requested changes; in the response I indicate in yellow the changed text and in the manuscript, I indicate in track changes. I made substantial changes that greatly improved the paper, thank you very much. If you have any further questions or if needed any additional changes, please just let me know best regards Tania

Reviewer 3

This is a very interesting study about an important issue that is often overlooked – the primary caregiver and his burden.

A few suggestions:

ABSTRACT

Please elaborate on the methods (a questionnaire study, sampling method).  

A convenience sample was used and data were collected via an online questionnaire.

INTRODUCTION

Lines 31-32 – non-communicable diseases and chronic diseases are the same.

has withdrawn non-communicable diseases

Lines 38-39 – this is not a clear sentence.

Thus, it can be seen that this increase of eldery the population and changes in the composition of households are also reflected in the number of informal caregivers

Lines 40-41 – please elaborate at least once on who is the informal caregiver (a definition will be good).

An informal caregivers is defined as the spouse or unmarried partner, relative or kin up to the 4th degree in the direct or collateral line of the person being cared for (e.g. children, grandchildren, great-grandchildren, great-great-grandchildren, siblings, parents, uncles, grandparents, great-grandparents, great-great-uncles or cousins) - Law No. 100/2019

Please add a short explanation in the INTRO about Law No. 100/2019. You refer to it in the manuscript, but it is not clear what exactly it means.

This law approves the Statute of the Informal Caregiver, which regulates the rights and duties of the caregiver and the person being cared for, establishing the respective support measures.

At the end of the INTRO, the study's aim is not clearly defined. Please make shorter sentences and try to make it clear and concise.

The main objective is to characterise and understand the difficulties experienced by informal caregivers in the context of Pandemic COVID-19 from a bio-psychosocial and environmental perspective.

METHODS

2.1 participants

Please mention here the sampling methods (convenient).

This is a cross-sectional, quantitative study, with a convenience sample.

2.3 procedure

Please elaborate on the statistical analysis. It is lacking.

Data analysis

As regards the statistical procedures, the analysis was performed using the Statistical Package for the Social Sciences (SPSS) version 25 for Windows. The descriptive statistics analysis of the instrument dimensions and the total scores of the instruments was per-formed, as well as the analysis of correlations and Student's t-test was used to analyse differences between groups. A linear regression model was calculated with the dependent variable of informal caregiver difficulties.

RESULTS

Please mention what was the time frame in which you collected the questionnaires.

Lines 256-260 – in the discussion, I would add an explanation to that point. Perhaps caregivers with less burden have a higher quality of life? The direction of the association should be discussed. (please add to the discussion, lines 347-349).

Higher levels of quality of life are negatively associated with perception of the caregiver's burden, perception of difficulties by the caregiver and higher levels of quality of life is positively associated with higher perception of social support, specifically with intimacy

Discussion

Lines 340-342 – not clear.

Perception of quality of life and its dimensions were considered as psychological variables and satisfaction with social support and its dimensions as social component. In this way, the analysis of the psychosocial variables under study revealed higher values in the dimensions of total, psychological and physical quality of life. In the satisfaction with social support, family and friends emerged as the most relevant.

Lines 372-376 – this is a very important sentence. But it is too long. Please rephrase and maybe make 2 sentences to make it easier for the reader to understand.

The sociodemographic characteristics of informal caregivers may also function as factors related to risk and protection. The results show that being a woman, being older, not having a professional activity and having higher education can lead to more difficulties. Focusing on the greatest difficulties experienced by caregivers with higher levels of education, particularly related to social problems and restrictions probably resulting from a greater conciliation of the informal caregiver's activity with the remaining family and professional obligations.

You need to add to the discussion a paragraph with the subheading: Strengths and limitations. Please add there that you do not have a response rate (you did not mention how many people received the questionnaire) and did not compare those who chose to answer and those who chose not to answer. This is a strong selection bias and should be addressed. 

The study presents some limitations, namely, reveals statistical analysis of some-what oversimplified data, it is an exploratory study, with very rich and original data, in the future it is intended to move to more specific and deeper analysis but it is considered an important contribution the results obtained. Since we have taken into account the new law on informal caregivers and this only covers informal family care-givers, we have chosen to only consider these. This may be considered a limitation as neighbours and the community also play an important role as informal caregiver and support. In a future study, we propose to expand the scope of informal caregivers to include other people who also have this function and are not included in this study. On the other hand, the present study used social support provided by family, friends, intimacy and social support activities. The study does not cover all forms of social support, this aspect can be complemented in qualitative studies through interviews with caregivers to understand if these have different impacts on people's psychological wellbeing, and how. Accessing the population of informal caregivers is not easy, in a pandemic context and with the use of an online questionnaire the task becomes even more difficult. We believe that the most accurate way to access the population would be through associations and formal networks of caregivers. As the dissemination of the questionnaire was made by entities related to caregivers, it was not possible for us to have a response rate since we do not know how many people received the questionnaire. Another limitation related to this aspect is that we were not able to compare those who chose to answer and those who chose not to answer. We believe that the sample obtained in the adverse context in which it took place is considerable and mitigates to some extent the above mentioned limitations.

Dear Reviewer The information on limitations and strength has been introduced in the conclusions section, if you think it should have a separate section we have included it but if it is possible to keep it like this we would like it more. Thanks

Reviewer 4 Report

I would like to thank the authors for the extensive analysis in this research project which was presented in the paper. However, the paper overall lacks coherence and hard to understand the aims and the methods. The way the paper presented is more of a thesis presentation type. 

Comment 1: The introduction is very lengthy and the study questions is not clear and the relation between Covid era and the study is not described anywhere in the manuscript. 

Comment 2: Is there a reason why results were described in the materials and methods section. 

Comment 3: The aim of the study doesn't have an relation to Covid-19 pandemic, using era as a proxy is okay but adding an analysis between covid and precovid is much needed to highlight the effect of pandemic on the burden of informal primary caregivers.

Comment 4: (L314-323) if the law was implemented in 2019 and the pandemic started in 2020 how would the affect the results of the current research project. Accurate description is warranted to give the paper scientific soundness. Response rate in a questionnaire doesn't reflect actual burden, this concept should've been acknowledged in the limitations in discussion section. 

Author Response

Revisions Paper

“Relationship between burden, quality of life and difficulties of informal primary caregivers in the context of the pandemic COVID-19: analysis of the contributions of public policies”

Dear Editor and Reviewer

I greatly appreciate the careful analysis and recommendations indicated by the reviewers. I have made the requested changes; in the response I indicate in yellow the changed text and in the manuscript, I indicate in track changes. I made substantial changes that greatly improved the paper, thank you very much. If you have any further questions or if needed any additional changes, please just let me know best regards Tania

Reviewers 4

Comment 1: The introduction is very lengthy and the study questions is not clear and the relation between Covid era and the study is not described anywhere in the manuscript. 

The COVID-19 pandemic, the respective isolation measures, difficulties in accessing health and social services that were focused exclusively on fighting the pandemic at the time, had a greater impact on certain groups, namely the elderly, people with chronic illnesses, people with socio-economic difficulties, women and the unemployed. Informal caregivers and their cared-for persons were one of the groups most affected by the pandemic and its restrictions [34, 35, 36].

As Liu et al. [37] revealed informal caregivers were extremely important in the early phase of the pandemic when people’s mobility and everyday activities have been con-siderably influenced by lockdown measures and other containment interventions. Vulnerable groups (such as older people and low-income people) were highly dependent on informal caregivers. And Informal caregivers have contributed to Covid containment and recovery.

The main objective is to characterise and understand the difficulties experienced by informal caregivers in the context of Pandemic COVID-19 from a bio-psychosocial and environmental perspective.  The difficulties experienced by informal caregivers was considered in terms of relational problems (RP), social constraints (SR), care demands (CE), reactions to caregiving (RC), family support (F) and professional support. We aimed to characterize these difficulties from a bio-psychosocial and environmental perspective, taking into account the socio-demographic and health characteristics of the informal caregiver and the person cared for, quality of life, perceived burden, social support, and the impact of the Covid-19 Pandemic on the informal caregiver and the person cared for. The following research questions can be asked: Will the informal caregiver's socio-demographic characteristics, health, quality of life, and social support influence his/her perception of difficulties as a caregiver? What supports and benefits were most used by caregivers in the context of pandemic COVID-19?

Comment 2: Is there a reason why results were described in the materials and methods section.

Tables 1 and 2 have been placed in the results section

Comment 3: The aim of the study doesn't have an relation to Covid-19 pandemic, using era as a proxy is okay but adding an analysis between covid and precovid is much needed to highlight the effect of pandemic on the burden of informal primary caregivers.

We indicate that the data collection was conducted in the pandemic context, we do not mention that it is the "impact" of the pandemic.  We have included questions in the questionnaire where we explicitly ask caregivers what has changed with the pandemic.

The Frequency of Responses analysis of the Covid-19 consequences in relation to caregivers and the person being cared for reveals that 81% of the informal caregivers reported more difficulty in accessing consultations and 65% in accessing services and support. There is also a worsening in the caregiver's anxiety levels (41%) and in their levels of worry (56%). There is an increase in the degree of isolation of the caregiver (69%) and the person being cared for (75%). Isolation and difficult access to health and other services led to increased symptomatology/needs/specificities of the individual being cared for according to 52% of the informal caregivers. A minority report that the pandemic has limited the time/contact with the individual cared for (15%) and that it has reduced the existence of affection with the individual cared for (17%). Line 384-393

Comment 4: (L314-323) if the law was implemented in 2019 and the pandemic started in 2020 how would the affect the results of the current research project.

 The law was passed in 2019, but it took a few months for the Ministry of Social Security to implement it.

Accurate description is warranted to give the paper scientific soundness. Response rate in a questionnaire doesn't reflect actual burden, this concept should've been acknowledged in the limitations in discussion section. 

The study presents some limitations, namely, reveals statistical analysis of somewhat oversimplified data, it is an exploratory study, with very rich and original data, in the future it is intended to move to more specific and deeper analysis but it is considered an important contribution the results obtained. Since we have taken into account the new law on informal caregivers and this only covers informal family caregivers, we have chosen to only consider these. This may be considered a limitation as neighbours and the community also play an important role as informal caregiver and support. In a future study, we propose to expand the scope of informal caregivers to include other people who also have this function and are not included in this study.

Accessing the population of informal caregivers is not easy, in a pandemic context and with the use of an online questionnaire the task becomes even more difficult. We believe that the most accurate way to access the population would be through associations and formal networks of caregivers. As the dissemination of the questionnaire was made by entities related to caregivers, it was not possible for us to have a response rate since we do not know how many people received the questionnaire. Another limitation related to this aspect is that we were not able to compare those who chose to answer and those who chose not to answer. We believe that the sample obtained in the adverse context in which it took place is considerable and mitigates to some extent the above mentioned limitations.

Round 2

Reviewer 1 Report

I'm ok with the revision.

Author Response

Revisions Paper

“Relationship between burden, quality of life and difficulties of informal primary caregivers in the context of the pandemic COVID-19: analysis of the contributions of public policies”

Dear Editor and Reviewer

I am very grateful for the feedback, and opportunity to improve and publish our paper.

Thanks very much best regards Tania

Reviewer 4 Report

I would like to thank the authors for this efforts in this research paper. The revised version is improved and it is well written. 

Here are my comments, 
The supplemental material document is a word file for the whole manuscript. Kindly provide the supplemental if you think it would help the reader understand the project more. 
The presentation of the manuscript and data is of a narrative type which blurs some important information and results that are already presented in the manuscript. 
Table 4&5 all the variables are significant, however the way it is presented in confusing. You should chose use either 0.1 or 0.5.  

Author Response

Revisions Paper

“Relationship between burden, quality of life and difficulties of informal primary caregivers in the context of the pandemic COVID-19: analysis of the contributions of public policies”

Dear Editor and Reviewer

I greatly appreciate the careful analysis. I Respond to comments below. If you have any further questions or if needed any additional changes, please just let me know best regards Tania

Reviewers 4

The supplemental material document is a word file for the whole manuscript. Kindly provide the supplemental if you think it would help the reader understand the project more. 
The presentation of the manuscript and data is of a narrative type which blurs some important information and results that are already presented in the manuscript. 

It was my mistake, I thought I should put the paper with the changes in track changes in supplementary material so that the reviewers could more easily confirm the changes.

Table 4&5 all the variables are significant, however the way it is presented in confusing. You should chose use either 0.1 or 0.5.  

In tables 4 & 5 I put ** when the significance is <0.05 and * when the significance is <0.01, I can change it if I consider it more appropriate, how should I put it to make it clearer?
